# Opioid Dose, Pain, and Recovery following Abdominal Surgery: A Retrospective Cohort Study

**DOI:** 10.3390/jcm11247320

**Published:** 2022-12-09

**Authors:** Dongxu Chen, Xiaoqing Li, Yu Chen, Huolin Zeng, Jin Liu, Qian Li

**Affiliations:** 1Department of Anesthesiology, West China Hospital, Sichuan University, No. 37 Wainan Guoxue Rd., Chengdu 610041, China; 2Department of Gastroenterology, West China Hospital, Sichuan University, No. 37 Wainan Guoxue Rd., Chengdu 610041, China; 3West China School of Nursing, Sichuan University, No. 37 Wainan Guoxue Rd., Chengdu 610041, China; 4Department of Anesthesiology, West China Fourth Hospital, Sichuan University, No. 18, Sect. 3 People South Road, Chengdu 610041, China

**Keywords:** opioid utilization, acute pain, pain relief, quality of recovery

## Abstract

**Background:** The optimal dosage for opioids given to patients after surgery for pain management remains controversial. We examined the association of higher post-surgical opioid use with pain relief and recovery. **Methods:** We retrospectively enrolled adult patients who underwent elective abdominal surgery at our hospital between August 2021 and April 2022. Patients were divided into the “high-intensity” or “low-intensity” groups based on their post-surgical opioid use. Generalized estimating equation models were used to assess the associations between pain scores at rest and during movement on days 1, 2, 3, and 5 after surgery as primary outcomes. The self-reported recovery and incidence of adverse events were analyzed as secondary outcomes. **Results:** Among the 1170 patients in the final analysis, 293 were in the high-intensity group. Patients in the high-intensity group received nearly double the amount of oral morphine equivalents per day compared to those in the low-intensity group (84.52 vs. 43.80), with a mean difference of 40.72 (95% confidence interval (CI0 38.96–42.48, *p* < 0.001) oral morphine equivalents per day. At all timepoints, the high-intensity group reported significantly higher pain scores at rest (difference in means 0.45; 95% CI, 0.32 to 0.58; *p* < 0.001) and during movement (difference in means 0.56; 95% CI, 0.41 to 0.71; *p* < 0.001) as well as significantly lower recovery scores (mean difference (MD) −8.65; 95% CI, −10.55 to −6.67; *p* < 0.001). A post hoc analysis found that patients with moderate to severe pain during movement were more likely to receive postoperative high-intensity opioid use. Furthermore, patients in the non-high-intensity group got out of bed sooner (MD 4.31 h; *p* = 0.001), required urine catheters for shorter periods of time (MD 12.26 h; *p* < 0.001), and were hospitalized for shorter periods (MD 1.17 days; *p* < 0.001). The high-intensity group was at a higher risk of chronic postsurgical pain (odds ratio 1.54; 95% CI, 1.14 to 2.08, *p* = 0.005). **Conclusions:** High-intensity opioid use after elective abdominal surgery may not be sufficient for improving pain management or the quality of recovery compared to non-high-intensity use. Our results strengthen the argument for a multimodal approach that does not rely so heavily on opioids.

## 1. Introduction

Acute pain occurs after almost every type of surgery [1]. The incidence of moderate to severe pain after abdominal surgery affects 42% of patients at rest but increases to 74% within the first 24 h after surgery among patients who engage in early mobilization [2,3]. Severe pain and inadequate analgesia can impair functional recovery, delay postoperative ambulation, and increase the risk of complications [1,4,5]. Therefore, the early control of acute postoperative pain is important.

Opioids are the primary pharmacological treatment for acute postoperative pain [1,6]. Most patients (72%) receive opioids for postoperative pain management [1], and this percentage exceeds 90% among patients experiencing severe pain [1,7]. In contrast to the idea that a greater opioid dose translates to better pain management, several studies suggest that high-intensity postoperative opioid use does not provide better analgesia than low-intensity use [8,9,10]. For example, patients who received a high-intensity dose (≥91 mg) of oral morphine equivalents (OME) at 1 day after lumbar spine surgery did not experience better pain relief or functional recovery than patients who received low-intensity (≤40 mg) or medium-intensity doses (41–90 mg) [11]. In addition, high-intensity opioid use is associated with longer hospital stays and lower-quality of recovery [12].

Determining the appropriate amount of opioids is complicated because of differences in typical pain management after various types of surgery and because patients can differ widely in pain thresholds. Studies have compared low- and high-intensity dose regimes, but the samples have been small, and follow-up studies are short, and they may not have involved truly high-intensity doses [7,13]. Thus, dose regimes that provide sufficient analgesic effect without compromising quality of recovery have not been adequately explored.

We hypothesized that high-intensity opioid use would not improve pain after abdominal surgery significantly better than non-high intensity opioids. Therefore, we addressed this question by comparing the self-reported postoperative pain and quality of recovery scores between patients subjected to high- or non-high intensity dose regimes after abdominal surgery at our medical center.

## 2. Materials and Methods

### 2.1. Study Design

This study was approved by the Biomedical Research Ethics Committee of West China Hospital (2022–863), which waived the requirement for written informed consent due to the retrospective nature of the study. The present study was conducted and reported in accordance with the guidelines of the STrengthening the Reporting of OBservational studies in Epidemiology (STROBE) [14].

This was a retrospective study of 1240 adult patients who underwent elective abdominal surgery under general anesthesia at the West China Hospital of Sichuan University from August 2021 to April 2022. We included patients >18 years of age who were classified with a physical status of I–III according to the American Society of Anesthesiologists (ASA)’s scale. Patients were excluded if they (1) lacked postoperative pain scores within 5 days after surgery, (2) were transferred to the intensive care unit after surgery, or (3) used more than 60 mg/day OME for 30 days prior to the operation (according to the FDA defines of an opioid-tolerant patient) [15].

### 2.2. Data Collection

The following sociodemographic and clinical data were extracted from electronic medical records: birth year, sex, height, and weight and a history of somatic diseases, ASA class, preoperative analgesic use, surgical information (i.e., the location and the type of surgery, duration, and intraoperative analgesic use), anesthesia method, and postoperative analgesia regimes. Pain intensity scores were assessed at postoperative days 1, 2, 3, and 5 at rest and with movement, using a numerical rating scale, and were collected from nurses’ documentation and averaged per post-operative day. The patients’ quality of recovery was obtained from the Quality of Recovery—15 questionnaire (QoR-15, ranging from 0 to 150; with a higher score indicating a better quality of recovery) [16], which is a patient-reported outcome measure evaluating recovery after surgery. The Chinese version of the QoR-15 (Appendix A Figure A1) has been validated in the Chinese population, and we have been authorized to use it. Furthermore, data on the length of hospital stay after surgery were also obtained from electronic medical records. The length of hospital stay after surgery was calculated as the number of days spent from the day of surgery to the day of discharge. The adverse events within 5 days after surgery were determined by medical record reviews.

### 2.3. OME Calculation

In the present study, postoperative opioids included sufentanil, fentanyl, pethidine, oxycodone, dezocine, tramadol, and hydromorphone. To be able to compare the intensity levels of postoperative prescriptions, we computed the OME (mg/day) by multiplying an opioid-specific conversion factor by the drug dose and quantity supplied [17]. The total OME was divided by the number of days (5 days) to obtain the average daily OME (calculation method is shown as below).

Calculation of OME daily dose (mg/day) = [Total opioid dose (within postoperative 5 days) × Conversion factor]/5 days

The average daily OME in the upper quartile of opioid utilization is defined as “high-intensity” opioid utilization [11,18].

### 2.4. Primary and Secondary Outcomes

The primary outcome was the acute pain scores as reported by the patient. Pain at rest and during movement was assessed at postoperative days 1, 2, 3, and 5 using a scale from 0 to 10, with 0 signifying “no pain” and 10 signifying “extreme pain” (mild, 1 to 3; moderate, 4 to 6; severe, 7 to 10).

Secondary outcomes included the quality of recovery, the incidence of adverse outcomes, and the incidence of chronic postsurgical pain (CPSP). The quality of recovery was assessed at postoperative day 5 using the patient-reported Quality of Recovery—15 questionnaire (QoR-15) [16]. Scores on the QoR-15 can range from 0 to 150, with higher scores indicating a better quality of recovery. The difference in the QoR-15 scores on the postoperative day 5 between the high-intensity and non-high intensity groups greater than 8 points was considered clinically meaningful [19]. Post-operation nausea and/or vomiting (PONV) and pruritus were considered as adverse events, and their incidence were recorded on postoperative days 1, 2, 3, and 5. We also compared key postoperative milestones: when patients got out of bed for the first time (hours), urine catheter retention time (hours), and the length of hospitalization (days). CPSP was determined using the criteria published by the International Association for the Study of Pain and was assessed at postoperative 3 months [20]. CPSP was collected with a telephone interview at 3 months after surgery. For 3 months follow-up assessments, we would attempt to contact the patient at least three times on 3 separate days.

### 2.5. Statistical Analysis

Continuous variables were reported as mean + standard deviation (SD) or as median and interquartile range (IQR), while categorical variables were expressed as number and percentage. Intergroup differences in continuous variables were assessed for significance using Student’s *t*-test or the Wilcoxon rank-sum test if the data were skewed. Qualitative data were compared using a chi-square test or Fisher’s exact test where appropriate.

For acute pain, numerical rating scale pain scores at rest and with movement at 1 day and 2, 3, and 5 days after surgery were compared between the groups using regression based on a generalized estimating equations approach, with exposure group and time (1, 2, 3, and 5 days after surgery) as fixed effects and the participant as a random effect. An unstructured correlation model was used. The fully adjusted marginal means of pain scores aggregated across all time points were reported separately by group, with contrast for the difference in groups and 95% confidence interval (CI). We further explored the associations of non-high- or high-intensity opioid use and moderate-to-severe pain using logistic regression models. In all models, we adjusted for age (as a continuous variable) and sex (men or women). Body mass index (BMI) was constructed from height and weight measured at the initial assessment center visit. Preoperative comorbidity, preoperative analgesic use, location of surgery (upper or lower abdominal), the type of surgery (open or laparoscopic), ASA physical class (I–II or >III), perioperative management (including anesthesia method, peripheral nerve block (yes or no), local infiltration (yes or no), the dosage of sufentanil and remifentanil), duration of surgery (as a continuous variable), and PCIA (yes or no) were extracted. The *p*-value was adjusted using Bonferroni adjustments (0.05/2) for two primary outcomes. A *p*-value less than 0.025 was considered significant for the primary outcome. Then, we conducted a post hoc analysis to evaluate risk factors for “high-intensity” use. Variables with *p* < 0.15 in the univariate analysis and the severity of postoperative pain were entered into multivariate logistic regression.

During the peer-review process, reviewers suggested that a group analysis on the type of surgery should be added due to the observation that patients who received high-intensity opioids were more likely to undergo open surgery than laparoscopic surgery. Three-way repeated measures repeated measures analysis of variance (ANOVA) was used to observe the effect of exposure (high- or non-high intensity opioid use) and the type of surgery (open or laparoscopic) on the outcome variable (pain score at postoperative days 1, 2, 3, and 5). In the model, the exposure groups and type of surgery were considered as between-subjects factors, whereas time (postoperative days 1, 2, 3, and 5) was considered as within-subjects factor.

As the quality of recovery, we examined the association between patients with “high-” versus “non-high-intensity” opioids use after surgery and the QoR-15 scores using median regression models, and the associations are represented as differences and their 95% CI. For other secondary outcomes, the association test was compared by logistic regression or median regression models with fully adjusted covariates.

There was no plan for the adjustment of the width of confidence intervals for multiple comparisons in the analyses of secondary outcomes. A *p*-value less than 0.05 in those outcomes should be interpreted as suggestive. All statistical hypothesis tests were two-sided. All the analyses were performed with R software, version 4.2.1 (The R Foundation for Statistical Computing; Sichuan, China; https://www.r-project.org/; accessed on 23 June 2022).

No statistical power calculation was conducted before the study. The sample size was based on the available data. We performed a post hoc analysis to assess the effect size that we could detect based on actual data. With our current sample size, we had 80% power at a two-sided alpha level of 0.05 to detect a greater than 0.4-point difference in numerical rating scale pain scores at rest or during movement on postoperative day 1, 2, 3, or 5 between the two groups.

## 3. Results

Of the 1240 patients retrospectively identified, 12 were excluded because their ASA classification was over III, 21 were excluded because they were transferred to the intensive care unit, and 37 were excluded due to inadequate data or loss to follow-up. The final analysis included 1170 patients (36.75% female) with a median age of 55 years (IQR 47–62; Figure 1).

One-quarter of patients (*n* = 293) received high-intensity opioids after surgery. On average, the high-intensity group received nearly double the daily OME as compared to the non-high-intensity group (84.52 vs. 43.80), corresponding to a mean difference (MD) of 40.72 mg/day OME (95% CI 38.96–42.48, *p* < 0.001).

Compared to patients who received non-high-intensity opioids, those who received high-intensity opioids were significantly older (56.00 vs. 55.00 year, *p* = 0.018) and more likely to undergo open surgery than laparoscopic surgery (80.21% vs. 68.30%, *p* < 0.001) in the lower abdomen (70.99% vs. 50.51%, *p* < 0.001; Table 1). They were also less likely to receive local infiltration (32.08% vs. 39.57%, *p* = 0.026), and their surgeries lasted longer (214.00 vs. 207.00 min, *p* = 0.029). Significantly more patients in the non-high-intensity group were treated with nonsteroidal anti-inflammatory agents (NSAIDs) on preoperative day 1 (23.95% vs. 1.37%, *p* < 0.001) and within 5 days after surgery. Multiple analgesics (i.e., two opioid) were administered for a greater proportion of high-intensity patients than non-high-intensity patients (80.89% vs. 48.78%; *p* < 0.001; Appendix A Figure A2).

### 3.1. Primary Outcome

Pain scores at rest and during movement were significantly worse in the high-intensity group than in the non-high-intensity group although the difference was small in clinical terms (Figure 2). The overall difference in pain scores at rest between the two groups was 0.45 (95% CI, 0.32 to 0.58; *p* < 0.001), with movement (differences in means, 0.56; 95% CI, 0.41 to 0.71; *p* < 0.001). Generalized estimation equations showed that patients in the high-intensity group were significantly more likely to report higher pain scores at 1, 2, 3, and 5 days after surgery even if the difference in pain was clinically small (Table 2). The high-intensity group showed a higher incidence of moderate-to-severe pain at rest or during movement (Appendix A Table A1).

The post hoc analysis found that patients with moderate to severe pain during movement on postoperative days 1, 2, 3, and 5 were more likely to receive high-intensity opioid use (Table 3). However, minimally invasive (OR 0.58; 95% CI, 0.41 to 0.81, *p* = 0.002) or NSAIDs administration at preoperative day 1 (OR 0.05; 95% CI, 0.02 to 0.12, *p* < 0.001) could reduce the risk of postoperative high-intensity opioid use.

A post hoc analysis suggested by a reviewer found that there was no interactions between exposure and the type of surgery on the pain score at rest (F = 2.09, *p* = 0.148; Appendix A Figure A3) and movement (F = 2.40, *p* = 0.121; Appendix A Figure A3), and non-significant interactions were observed for the three-way of interactions of time, the type of surgery groups, and exposure groups.

### 3.2. Secondary Outcomes

Based on QoR-15 scores, patients in the non-high-intensity group reported better recovery than those in the high-intensity group on postoperative day 5 (126.00 vs. 105.00). The difference after adjusting for potential confounders was an MD of −8.65 (95% CI, −10.55 to −6.67, *p* < 0.001), which was clinically significant (Figure 3).

Patients in the high-intensity group took longer to get out of bed after surgery (MD 4.31 h; 95% CI, 2.81 to 6.48, *p* = 0.001), a difference that was clinically small (Table 4). These individuals also required a longer use of urine catheters (MD 12.26 h; 95% CI, 8.97 to 16.95, *p* < 0.001) and longer hospitalization after surgery (MD 1.17 days; 95% CI, 1.06 to 2.07, *p* < 0.001). CPSP occurred in 113 of 293 patients (38.57%) that were given high-intensity opioids compared to 240 of 877 patients (27.37%) given non-high-intensity opioids. These proportions were significantly different after adjusting for potential confounders (OR 1.54; 95% CI, 1.14 to 2.08, *p* = 0.005). In contrast, the two groups did not differ significantly in the incidence of PONV or pruritus.

## 4. Discussion

Postoperative management of acute pain is challenging, as it requires balancing the need to ensure adequate analgesia and the need to minimize adverse effects, particularly overdosing [4]. Our findings suggest that high-intensity opioid use is associated with a lower quality of early recovery and similar or worse pain relief than non-high-intensity opioid use. In addition, postoperative pain intensity and postoperative opioid consumption are not independent of each other but interlinked, and poorly controlled pain during movement is associated with an increased risk of postoperative high-intensity opioid use. These outcomes indicate that using multimodal approaches to improve pain resolution and to reduce postoperative opioid consumptions remain a priority.

Pain management varies widely based on physicians’ preferences, institutional guidelines, and patient characteristics. Opioid doses ranging from 0 to 555 OME (mg/day) have been used for patients following orthopedic surgery [12]. Our study also showed wide variability in the opioid dose (84.52 vs. 43.80 mg/day OME), but higher doses did not translate to better pain relief. In fact, the mean pain score at 5 days after surgery was still 3 points greater among patients receiving high-intensity opioids. Similarly, another study reported that a high dose of opioids on postoperative day 1 did not improve analgesia during the subsequent 24 h in older adults who underwent non-cardiac surgery [13]. Our analysis involved a follow-up study for five days after surgery, which may render our results more reliable than those based only on a follow-up study for 1 postoperative day [7,13].

Opioids are one of the most effective analgesics for moderate to severe post-operative pain [21]. However, a high dose of opioids is associated with side effects such as PONV, respiratory depression, and immune system depression [22]. High doses of opioids for acute postoperative pain management might impede postoperative recovery and have long-term consequences [23]. However, the study reported that nearly all opioid-naive patients following spine surgery discontinued opioid use 6 months after surgery and did not experience opioid dependency [24]. However, the latest finding is that a higher perioperative opioid prescription is associated with long-term (postoperative days 91–365) opioid utilization, which indicates an increased the risk of opioid dependence [18]. Indeed, we found in our sample that the risk of CPSP was higher with higher opioid dosages. As a result, the long-term consequences of “high-intensity” opioid use in the perioperative period require further research.

Despite the common use of opioids for managing postoperative pain, clinical guidelines with opioid-sparing measures are emerging [25,26]. Patients who do not benefit from high-intensity opioids may benefit from multimodal approaches or more aggressive rehabilitation or interdisciplinary pain management [4,27]. Patients receiving low-intensity opioids were more likely to receive nonsteroidal anti-inflammatory agents, which may help explain their lower pain. This strengthens the argument for a multimodal approach that does not rely so heavily on opioids, which was recommended in a recent consensus statement from the American Society for Enhanced Recovery and Perioperative Quality [28]. Our previous work with patients undergoing major abdominal surgery [29] showed that a multimodal analgesia approach reduced acute pain and promoted early recovery more effectively than a traditional strategy based on opioids alone. Large comparative studies of multimodal analgesic protocols that examine short- and long-term patient outcomes are required in order to standardize the number of opioids that can be prescribed.

We acknowledge several limitations in our study. First, the difference in postoperative opioids utilization for patients receiving treatment from “high-intensity”—roughly 40.72 OME per day during a 5-day period, which corresponds to approximately one and one-third morphine tablets per day—may not indicate that the patient with high doses of opioid use had been receiving doses that were sufficiently large enough to place the patient at a poorer quality of recovery and severe pain. Second, our study included only patients who underwent elective abdominal surgery, which may limit the generalizability of our results. Moreover, in the current study, the median age of patients was 55 years (IQR, 47–62), and more patients received open surgeries instead of laparoscopic surgeries; this may represent a selective bias, and our results need to be interpreted with caution. Third, we assessed pain levels and recovery only until 5 days after surgery, and we did not analyze additional outcomes within the first 24 h, so future studies should address these limitations in order to verify and extend our findings. Furthermore, postoperative pain was measured by a self-reported numeric rating scale in the current study. Using objective measurements is warranted to validate or refute our findings. For example, utilizing the bioimpedance methodology from the three-dimensional exploration of nociception [30] or the nociception level index (a multi-parameter technology for pain assessment) [31] to personalize the severity of pain is warranted. Future work should also use a prospective design to explore whether high opioid doses causally affect pain or recovery.

## 5. Conclusions

High-intensity opioid use after elective abdominal surgery may not be sufficient for improving pain outcomes or recovery compared to non-high-intensity use. Future research should explore opioid prescriptions across a range of surgeries and quantify the thresholds of overuse in order to help inform clinical guidelines for postoperative pain management. Less may be better, but the jury is still out.

## Figures and Tables

**Figure 1 jcm-11-07320-f001:**
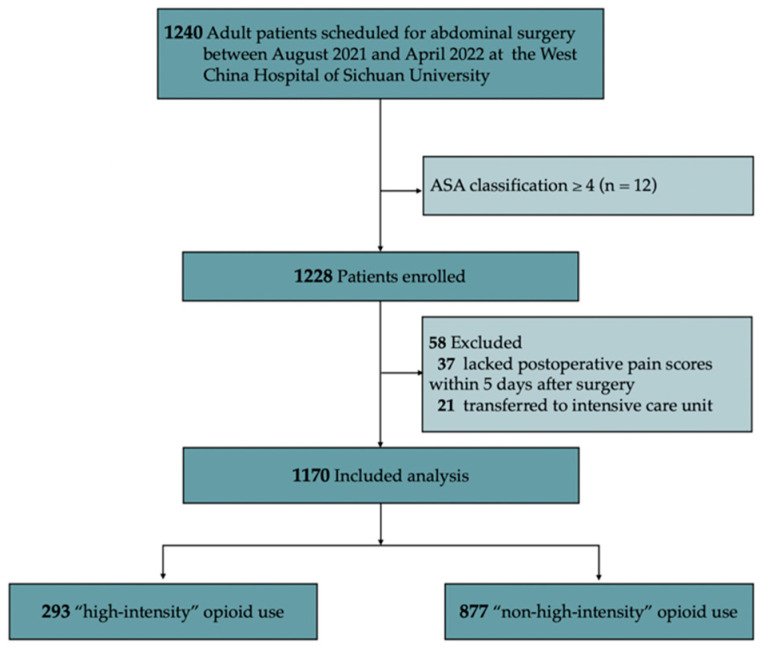
Flow chart of patient enrollment. Abbreviations: ASA, American Society of Anesthesiologists.

**Figure 2 jcm-11-07320-f002:**
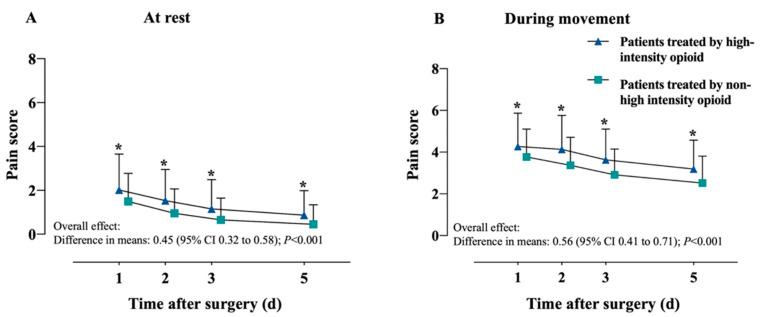
Average postoperative pain scores (**A**) during movement or (**B**) at rest. * *p* < 0.001. Abbreviations: CI, confidence interval. Cyan represents patients treat by “high-intensity” opioid; Light green represents patients treated by “non-high-intensity” opioid.

**Figure 3 jcm-11-07320-f003:**
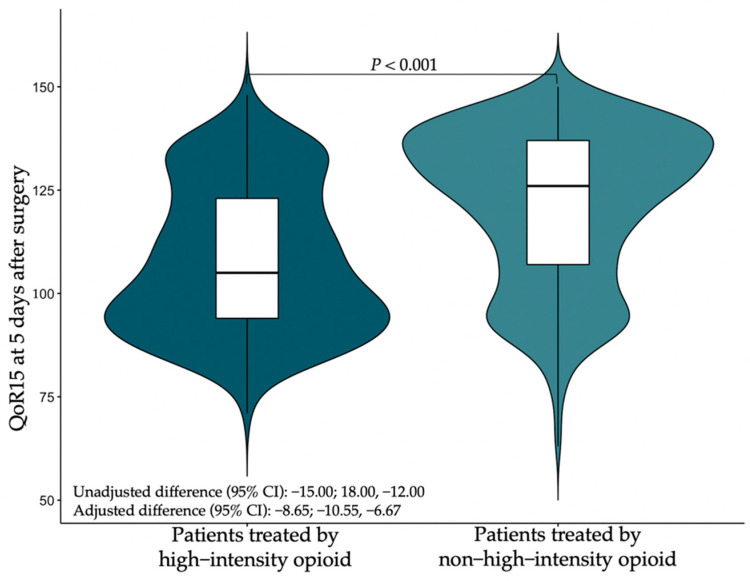
Violin plot of patient QoR-15 scores at 5 days after surgery. Abbreviations: QoR-15, Quality of Recovery—15; CI, confidence interval. Cyan represents patients treat by “high-intensity” opioid; Light cyan represents patients treated by “non-high-intensity” opioid.

**Table 1 jcm-11-07320-t001:** Clinicodemographic and clinical characteristics of patients given opioids after abdominal surgery.

Characteristics	Total *n* = 1170	Patients Treated by High-Intensity Opioid *n* = 293	Patients Treated by Non-High-Intensity Opioid *n* = 877	*p*
Age, years, median (IQR)	55.00 (47.00, 62.00)	56.000 (49.00, 64.00)	55.00 (47.00, 62.00)	0.018 ^a^
Age group, *n* (%)				0.092 ^b^
<65 years	955 (81.62)	229 (78.16)	726 (82.78)	
≥65 years	215 (18.38)	64 (21.84)	151 (17.22)	
Sex, *n* (%)				0.979 ^b^
Male	740 (63.25)	186 (63.48)	554 (63.17)	
Female	430 (36.75)	107 (36.52)	323 (36.83)	
BMI, kg/m^2^, median (IQR)	22.60 (20.55, 24.81)	22.94 (20.76, 25.09)	22.49 (20.52, 24.80)	0.288 ^a^
BMI group, *n* (%)				0.934 ^b^
<24	771 (65.90)	192 (65.53)	579 (66.02)	
≥24	399 (34.10)	101 (34.47)	298 (33.98)	
Preoperative comorbidities, *n* (%)				
Hypertension	164 (14.02)	41 (13.99)	123 (14.03)	>0.999 ^b^
Diabetes mellitus	104 (8.89)	23 (7.85)	81 (9.24)	0.546 ^b^
Cardiovascular disease	8 (0.68)	2 (0.68)	6 (0.68)	>0.999 ^c^
COPD	44 (3.76)	20 (6.83)	24 (2.74)	0.003 ^b^
Location of surgery, *n* (%)				<0.001 ^b^
Lower abdominal	651 (55.64)	208 (70.99)	443 (50.51)	
Upper abdominal	519 (44.36)	85 (29.01)	434 (49.49)	
Type of surgery, *n* (%)				<0.001 ^b^
Open	834 (71.28)	235 (80.21)	599 (68.30)	
Laparoscopic	336 (28.72)	58 (19.79)	278 (31.70)	
Anesthesia method, *n* (%)				0.180 ^b^
Inhalation or total intravenous anesthesia	821 (70.17)	196 (66.89)	625 (71.27)	
Inhalation and intravenous anesthesia	349 (29.83)	97 (33.11)	252 (28.73)	
ASA class, *n* (%)				0.906 ^b^
I~II	967 (82.65)	241 (82.25)	726 (82.78)	
III	203 (17.35)	52 (17.75)	151 (17.22)	
NSAIDs use at preoperative day 1, *n* (%)	214 (18.29)	4 (1.37)	210 (23.95)	<0.001 ^c^
Opioid use at preoperative day 1, *n* (%)	14 (1.20)	5 (1.71)	9 (1.03)	0.358 ^c^
Intraoperative sufentanil, μg, median (IQR)	32.50 (27.50, 40.00)	35.00 (27.50, 40.00)	32.50 (27.50, 38.00)	0.098 ^b^
Intraoperative remifentanil, μg, median (IQR)	1751.95 (1248.23, 2323.75)	1825.00 (1428.00, 2259.00)	1723.00 (1226.00, 2328.00)	0.020 ^b^
Duration of surgery, min, median (IQR)	210.00 (160.00, 271.00)	214.00 (175.00, 269.00)	207.00 (152.00, 272.00)	0.029 ^a^
Duration of surgery group, *n* (%)				0.487 ^b^
<2 h	82 (7.01)	16 (5.46)	66 (7.52)	
2~4 h	670 (57.26)	171 (58.36)	499 (56.90)	
≥4 h	418 (35.73)	106 (36.18)	312 (35.58)	
Local infiltration, *n* (%)	441 (37.69)	94 (32.08)	347 (39.57)	0.026 ^b^
Peripheral nerve block, *n* (%)	752 (64.27)	199 (67.92)	553 (63.06)	0.152 ^b^
PCIA, *n* (%)	992 (84.79)	244 (83.28)	748 (85.29)	0.461 ^b^
NSAIDs use on postoperative day, *n* (%)				
1	261 (22.31)	30 (10.24)	231 (26.34)	<0.001 ^b^
2	261 (22.31)	33 (11.26)	228 (26.00)	<0.001 ^b^
3	228 (19.49)	32 (10.92)	196 (22.35)	<0.001 ^b^
4	211 (18.03)	31 (10.58)	180 (20.53)	<0.001 ^b^
5	187 (15.98)	29 (9.90)	158 (18.02)	<0.001 ^b^

Data presented as *n* (%), mean ± SD, or median (quartile 1, quartile 3), unless otherwise indicated; ^a^ Student’s *t*-test; ^b^ chi-square test; ^c^ Fisher’s exact text; Abbreviations: ASA, American Society of Anesthesiologists; BMI, body mass index; COPD, chronic obstructive pulmonary disease; IQR, interquartile range; NSAID, nonsteroidal anti-inflammatory agent; PCIA, patient-controlled intravenous analgesia.

**Table 2 jcm-11-07320-t002:** Generalized estimating equation model to identify associations between intensity of postoperative opioid use and pain scores at rest and during movement.

Pain Scores	Patients Treated by High-Intensity Opioid *n* = 293	Patients Treated by Non-High-Intensity Opioid *n* = 877	Unadjusted Differences (95% CI)	*p*	Adjusted Differences (95% CI) **	*p*
At rest						
Model effect *						
Overall effect, marginal mean	1.77 (1.36)	1.25 (1.44)	0.51 (0.42, 0.59)	<0.001	0.45 (0.32, 0.58)	<0.001
Time				<0.001		<0.001
Time × exposure interaction †				0.410		0.514
1 day, mean ± SD	2.02 (1.63)	1.49 (1.28)	0.53 (0.32, 0.73)	<0.001	0.47 (0.26, 0.67)	<0.001
2 days, mean ± SD	1.53 (1.42)	0.95 (1.11)	0.57 (0.40, 0.75)	<0.001	0.51 (0.33, 0.69)	<0.001
3 days, mean ± SD	1.15 (1.34)	0.65 (1.00)	0.50 (0.33, 0.66)	<0.001	0.37 (0.22, 0.51)	<0.001
5 days, mean ± SD	0.87 (1.12)	0.45 (0.89)	0.42 (0.28, 0.56)	<0.001	0.44 (0.27, 0.61)	<0.001
During movement						
Model effect *						
Overall effect, marginal mean	3.43 (1.36)	2.77 (1.23)	0.66 (0.57, 0.76)	<0.001	0.56 (0.41, 0.71)	<0.001
Time				<0.001		<0.001
Time × exposure interaction †				0.310		0.060
1 day, mean ± SD	4.27 (1.60)	3.77 (1.33)	0.50 (0.30, 0.71)	<0.001	0.40 (0.20, 0.61)	<0.001
2 days, mean ± SD	4.13 (1.63)	3.37 (1.34)	0.77 (0.56, 0.97)	<0.001	0.66 (0.45, 0.87)	<0.001
3 days, mean ± SD	3.63 (1.47)	2.91 (1.24)	0.71 (0.53, 0.90)	<0.001	0.61 (0.42, 0.79)	<0.001
5 days, mean ± SD	3.19 (1.38)	2.52 (1.29)	0.67 (0.49, 0.85)	<0.001	0.57 (0.39, 0.75)	<0.001

Numerical rating scale score: 0, no pain; 10, worst pain imaginable. * A generalized estimating equations model was run to compare numerical rating scale score differences between groups, with patients treated by high-intensity opioid in surgical ward (exposure) group and time (1, 2, 3, and 5 days after surgery) as fixed effects and participant as a clustering variable. † A generalized estimating equations model was run to compare numerical rating scale score differences between groups, with patients treated by high-intensity opioid in surgical ward (exposure) group and time (1, 2, 3, and 5 days after surgery) and exposure group × time interaction as fixed effects and participant as a clustering variable. ** Generalized estimating equations models were adjusted for age, sex, body mass index, preoperative analgesic and comorbidities, surgical information (i.e., the type of surgery, location of surgery, duration of surgery, anesthesia method, peripheral nerve block, local infiltration, intraoperative analgesic), and postoperative analgesic (i.e., patient-controlled intravenous analgesia or nonsteroidal anti-inflammatory agents). Abbreviations: SD, standard deviations; CI, confidence interval.

**Table 3 jcm-11-07320-t003:** Univariable and multivariable results based on the logistic model for identifying risk factors for “high-intensity” opioid use.

Variables	Multivariate Mode		Final Multivariate Model	
	OR (95% CI)	*p*-Value	OR (95% CI)	*p*-Value
**Age (years)**	1.01 (1.00, 1.02)	0.142		
**COPD** **(yes vs. no)**	1.51 (0.75, 2.97)	0.240		
**Location of surgery**				
Lower abdominal	ref			
Upper abdominal	0.77 (0.55, 1.08)	0.134		
**Type of surgery**				
Open	ref		ref	
Laparoscopic	0.61 (0.42, 0.87)	0.007	0.58 (0.41, 0.81)	0.002
**NSAIDs use at preoperative day 1 (yes vs. no)**	0.07 (0.02, 0.17)	<0.001	0.05 (0.02, 0.12)	<0.001
**Intraoperative remifentanil**	1.00 (1.00, 1.00)	0.219		
**Duration of surgery**	1.00 (0.99, 1.00)	0.264		
**Local infiltration (yes vs. no)**	0.92 (0.66, 1.28)	0.611		
**NSAIDs use on postoperative day (yes vs. no)**				
1	0.61 (0.22, 1.63)	0.337		
2	0.87 (0.30, 2.45)	0.789		
3	1.00 (0.09, 7.55)	0.997		
4	0.96 (0.20, 7.11)	0.964		
5	1.53 (0.48, 6.03)	0.501		
**Postoperative moderate-to-severe pain * at rest (yes vs. no)**				
Day 1	1.42 (0.91, 2.19)	0.119		
Day 2	1.25 (0.68, 2.29)	0.473		
Day 3	1.06 (0.48, 2.37)	0.880		
Day 5	0.80 (0.32, 2.00)	0.641		
**Postoperative moderate-to-severe pain * during movement (yes vs. no)**				
Day 1	1.40 (0.98, 2.00)	0.061	1.44 (1.02, 2.03)	0.036
Day 2	1.78 (1.23, 2.58)	0.002	1.89 (1.32, 2.73)	0.001
Day 3	1.49 (1.04, 2.12)	0.028	1.60 (1.14, 2.26)	0.007
Day 5	2.00 (1.42, 2.81)	<0.001	2.07 (1.49, 2.86)	<0.001

* Numeric rating scale more than 3 points. Abbreviations: COPD, chronic obstructive pulmonary disease; NSAID, nonsteroidal anti-inflammatory agent.

**Table 4 jcm-11-07320-t004:** Regression analyses to identify associations between intensity of postoperative opioid use and secondary outcomes.

Outcome and Postoperative Day	Patients Treated by High-Intensity Opioid *n* = 293	Patients Treated by Non-High-Intensity Opioid *n* = 877	Crude OR/Median Difference (95% CI)	Adjusted OR/Coefficient ^#^ (95% CI)	*p*
PONV after surgery, *n* (%)					
Day 1	31 (10.58)	168 (19.16)	0.50 (0.33, 0.74)	0.50 (0.31, 1.02)	0.052
Day 2	18 (6.14)	75 (8.55)	0.70 (0.40, 1.18)	0.58 (0.31, 1.04)	0.075
Day 3	11 (3.75)	42 (4.79)	0.78 (0.38, 1.49)	0.61 (0.27, 1.29)	0.211
Day 5	30 (10.24)	41 (4.68)	2.33(1.41, 3.80)	1.36 (0.79, 2.33)	0.263
Pruritus after surgery, *n* (%)					
Day 1	11 (3.75)	66 (7.53)	0.49 (0.24, 0.90)	0.77 (0.36, 1.54)	0.478
Day 2	5 (1.71)	31 (3.54)	0.49 (0.16, 1.16)	0.66 (0.21, 1.76)	0.437
Day 3	4 (1.37)	8 (0.91)	1.53 (0.39, 5.01)	-^&^	-^&^
Day 5	6 (2.05)	4 (0.46)	4.85 (1.32 18.40)	2.94 (0.73, 11.06)	0.141
Time to first get out of bed, h, median (IQR)	46.33 (40.00; 63.00)	41.00 (24.00; 50.00)	8.83 (6.00, 12.00)	4.31 (2.81, 6.48) *	0.001
Urine catheter retention time, h, median (IQR)	67.12 (43.22; 90.52)	41.00 (20.00; 64.00)	24.88 (22.08, 27.42)	12.26 (8.97, 16.95) *	<0.001
Length of stay after surgery, d, median (IQR)	8.50 (5.00; 8.50)	5.10 (3.30; 7.50)	2.00 (1.60, 2.30)	1.17 (1.06, 2.07) *	<0.001
Chronic postsurgical pain, *n* (%)	113 (38.57)	240 (27.37)	1.67 (1.26, 2.20)	1.54 (1.14, 2.08)	0.005

Data are *n* (%) or median (quartile 1, quartile 3) unless otherwise indicated. ^&^ The model did not converge. * Median regression. ^#^ Adjusted for age, sex, body mass index, preoperative analgesic and comorbidities, surgical information (type, location, and duration of surgery; anesthesia method; peripheral nerve block; local infiltration; intraoperative analgesic), and postoperative analgesia (patient-controlled intravenous analgesia or nonsteroidal anti-inflammatory agents). Abbreviations: CI, confidence interval; IQR, interquartile range; OR, odds ratio.

## Data Availability

The data used to support the findings of this study are included within the article and Appendix A. Further inquiries can be directed to the corresponding authors.

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
