# Peer review of "Opioid Dose, Pain, and Recovery following Abdominal Surgery: A Retrospective Cohort Study"

_jcm, 2022, doi:10.3390/jcm11247320_

Round 1

Reviewer 1 Report

1/ What is the influence/ impact of using a subjective measurement tool (self-report) on the results? Several monitoring devices are in research for objective pain management; a discussion should be added (please check the latest literature https://doi.org/10.3390/jcm9030684,  https://doi.org/10.3390/jcm9030684, https://doi.org/10.2147/JPR.S332845)

2/ How the age of the patients in the high-intensity opioids group may represent a bias on the results? Moreover, as they underwent the open surgery instead of a laparoscopic.  Why the authors motivate that the difference in age was not clinically significant? A group analysis on the type of surgery should be included.

3/ The patients are divided in 2 groups based only on the amount of opioids given after surgery. However, it misses from the study the information of the patient's pain level immediately after the surgery. Why the differences in Fig2 and Table 2 may not be considered to have the source in the surgery type or in the patient's self-reported pain?

Author Response

Manuscript No.: jcm-1949625

Title: Association of High Opioid Dose with Pain and Recovery Following Abdominal Surgery: A Retrospective Cohort Study

Dear Sophia Zhou,

Thank you for considering our manuscript for publication and kindly providing us the opportunity to revise the paper. We highly appreciate the insightful and critical comments from the reviewers and the editors. The manuscript has been carefully revised in light of those comments, changes have been highlighted in red, and a point-by-point response to each comment has been provided below. Our manuscript has been edited by MDPI (ID english-55183) to improve the general language; an editing certificate is available upon request. We are now submitting the revised manuscript for further consideration for publication in Journal of Clinical Medicine.

Should you have any comments or questions, please contact me at the address provided below.

Yours sincerely,

Qian Li, MD

Department of Anesthesiology

West China Hospital, Sichuan University

E-mail: hxliqian@foxmail.com

Response to Reviewers

Reviewer-1

Comment 1: What is the influence/ impact of using a subjective measurement tool (self-report) on the results? Several monitoring devices are in research for objective pain management; a discussion should be added (please check the latest literature https://doi.org/10.3390/jcm9030684, https://doi.org/10.2147/JPR.S332845).

Response: Thank you very much for your valuable advice. We have added a discussion as your suggestion (Page 10, line 338-343). Also, it was copied as below:

Furthermore, postoperative pain was measured by a self-reported numeric rating scale in the current study. Using objective measurements is warranted to validate or refute our findings. For example, utilizing the bioimpedance methodology from the three-dimensional exploration of nociception [30] or the nociception level index (a multi-parameter technology for pain assessment) [31] to personalize the severity of pain is warranted.

Reference:

[30] Neckebroek M, Ghita M, Ghita M, Copot D, Ionescu CM. Pain detection with bioimpedance methodology from 3-dimensional exploration of nociception in a postoperative observational trial. J Clin Med 2020;9(3):684. https://doi.org/10.3390/jcm9030684.

[31] Gélinas C, Shahiri T S, Richard-Lalonde M, Laporta D, Morin JF, Boitor M, et al. Exploration of a multi-parameter technology for pain assessment in postoperative patients after cardiac surgery in the intensive care unit: The nociception level index (NOL)TM. J Pain Res 2021;14:3723–31. https://doi.org/10.2147/JPR.S332845.

Comment 2: How the age of the patients in the high-intensity opioids group may represent a bias on the results? Moreover, as they underwent the open surgery instead of a laparoscopic. Why the authors motivate that the difference in age was not clinically significant? A group analysis on the type of surgery should be included.

Response: Many thanks for your comment.

(1) In current study, the median age was 55 years (IQR, 47-62) and more patients received open surgery instead of a laparoscopic, it may represent a selective bias on the results. We point out as a limitation in the revised Discussion (Page 10, lines 332-335). And it was copied as follow:

“In the current study, the median age of patients was 55 years (IQR, 47-62) and more patients received open surgeries instead of laparoscopic surgeries, this may represent a selective bias, and our results need to be interpreted with caution.”

(2) We have deleted the sentence about “the difference in age was not clinically significant”.

(3) For surgery type, we have added a group analysis as your suggestion. And it was copied as follow:

Method (Page 4, line 147-154): During the peer-review process, reviewers suggested that a group analysis on the type of surgery should be added due to the observation that patients who received high-intensity opioids were more likely to undergo open surgery than laparoscopic surgery. Three-way repeated measures repeated measures analysis of variance (ANOVA) was used to observe the effect of exposure (high or non-high intensity opioid use) and the type of surgery (open or laparoscopic) on the outcome variable (pain score at postoperative days 1, 2, 3, and 5). In the model, the exposure groups and type of surgery were considered as between-subjects factors, whereas time (postoperative days 1, 2, 3, and 5) was considered as within-subjects factor.

Result (Page 7, line 241-247): A post hoc analysis suggested by a reviewer found that there was no interactions between exposure and the type of surgery on the pain score at rest (F = 2.09, p = 0.148; Appendix Figure C); movement (F = 2.40, p = 0.121; Appendix Figure C), non-significant interactions were observed for the three-way of interactions of time, the type of surgery groups and exposure groups.

Comment 3: The patients are divided in 2 groups based only on the amount of opioids given after surgery. However, it misses from the study the information of the patient's pain level immediately after the surgery. Why the differences in Fig2 and Table 2 may not be considered to have the source in the surgery type or in the patient's self-reported pain?

Response: Many thanks for your comment.

For surgery type, as stated in your comment 2, patients who received high-intensity opioids more likely to undergo open than laparoscopic surgery, do not fully exclude the differences in Fig2 and Table 2 may result by the surgery type. We conducted a three-way repeated measures repeated measures analysis of variance (ANOVA) to observe the effect of exposure (high or non-high intensity opioid use) and type of surgery (open or laparoscopic) on the outcome variable (pain score at postoperative days 1, 2, 3, and 5). Results found that the surgery type have no effect on the association between exposure (high or non-high intensity opioid use) and outcome (pain score at postoperative days 1, 2, 3, and 5).

For patient’s self-reported pain, as stated in your comment 1, we have added a discussion as your suggestion (Page 10, line 338-343).

Finally, there was indeed no information of the patient’s pain level immediately after the surgery, we point out as a limitation in the revised Discussion (Page 10, lines 335-338).

“Third, we assessed pain levels and recovery only until 5 days after surgery, and did not analyze additional outcomes within the first 24 hours, so future studies should address these limitations in order to verify and extend our findings.”

Reviewer 2 Report

  1. There are differences between exclusion of figure1 and exclusion criteria. (incomplete data, ICU, lost f/u and died vs. (1) lacked postoperative pain scores within 5 days after surgery, (2) were transferred to the intensive care unit after surgery, or (3) used more than 60 mg/day OME for 30 days prior to the operation. )

  2. Why did you decide to exclude OME over 60 mg/day for 30 days in exclusion criteria? Are there references?

  3. How did you calculate OME? There is no detailed method calculating OME in your reference. ([16] Zeremski M, Zavala R, Dimova RB, Chen Y, Kritz S, Sylvester C, et al. Improvements in HCV-related knowledge among substance users on opioid agonist therapy after an educational intervention. J Addict Med 2016;10:104–9. https://doi.org/10.1097/ADM.0000000000000196.) I think it would be good to show the calculation method.

  4. You used QoR-15 to evaluate patients’ quality of recovery. The QoR form in the proposed reference was made in English. How did you use the QoR form? Is there an authorized version of your native language version form? If you didn't use the English form, it would be nice to show the form you used.

  5. In line 105, What is the minimum difference? Did you investigate the QoR several times?

  6. Did all patients visit the hospital 3 months after the surgery? Considering that there are not many excluded patients, I think it would be difficult to investigate for most patients pain score at 3 months after surgery.

  7. How did you define severe pain in table 3?

Author Response

Manuscript No.: jcm-1949625

Title: Association of High Opioid Dose with Pain and Recovery Following Abdominal Surgery: A Retrospective Cohort Study

Dear Sophia Zhou,

Thank you for considering our manuscript for publication and kindly providing us the opportunity to revise the paper. We highly appreciate the insightful and critical comments from the reviewers and the editors. The manuscript has been carefully revised in light of those comments, changes have been highlighted in red, and a point-by-point response to each comment has been provided below. Our manuscript has been edited by MDPI (ID english-55183) to improve the general language; an editing certificate is available upon request. We are now submitting the revised manuscript for further consideration for publication in Journal of Clinical Medicine.

Should you have any comments or questions, please contact me at the address provided below.

Yours sincerely,

Qian Li, MD

Department of Anesthesiology

West China Hospital, Sichuan University

E-mail: hxliqian@foxmail.com

Response to Reviewers

Reviewer-2

Comment 1: There are differences between exclusion of figure1 and exclusion criteria. (incomplete data, ICU, lost f/u and died vs. (1) lacked postoperative pain scores within 5 days after surgery, (2) were transferred to the intensive care unit after surgery, or (3) used more than 60 mg/day OME for 30 days prior to the operation. ).

Response: We apologize for the negligence in preparing the original manuscript. Indeed, we followed the exclusion criteria stated in the method section. The exclusion criteria as appearing in the original manuscript reflect practical issues that represent barriers to outcome assessment, but not related to the study design otherwise. Per reviewer comment, we have revised the patient flow chart.

Comment 2: Why did you decide to exclude OME over 60 mg/day for 30 days in exclusion criteria? Are there references?

Response: Such an exclusion criterion is consistent with FDA defines (https://www.fda.gov/drugs/information-drug-class/opioid-medications) and the latest published article (DOI:10.1097/10.1136/rapm-2021-10345). OME over 60 mg/day for 30 days represents chronic opioid use, which it may affect the amount of opioid use and pain sensitivity after surgery. We have provided the supporting references.

Reference:

“Nafziger AN, Barkin RL. Opioid therapy in acute and chronic pain. J Clin Pharmacol 2018;58(9):1111–22. https://doi.org/10.1002/jcph.1276.”

Comment 3: How did you calculate OME? There is no detailed method calculating OME in your reference. ([16] Zeremski M, Zavala R, Dimova RB, Chen Y, Kritz S, Sylvester C, et al. Improvements in HCV-related knowledge among substance users on opioid agonist therapy after an educational intervention. J Addict Med 2016;10:104–9. https://doi.org/10.1097/ADM.0000000000000196.) I think it would be good to show the calculation method.

Response: Thank you very much for your valuable advice. We have provided more detailed method calculating OME (Page 3, line 95-98). And it was copied as follow:

To be able to compare intensity levels of postoperative prescriptions, we computed the OME (mg/day) by multiplying an opioid-specific conversion factor by the drug dose and quantity supplied [16]. The total OME was divided by the number of days (5 days) to obtain the average daily OME (calculation method is shown as below).

Calculation of OME daily dose (mg/day) = [Total opioid dose (within postoperative 5 days) Í Conversion factor] / 5 days

Comment 4: You used QoR-15 to evaluate patients’ quality of recovery. The QoR form in the proposed reference was made in English. How did you use the QoR form? Is there an authorized version of your native language version form? If you didn't use the English form, it would be nice to show the form you used.

Response: Many thanks for your comment. The Chinese version of the QoR-15 has been validated in the Chinese population by our hospital’s research team (Xia Yun Zuo, Department of Anesthesiology, West China Hospital, Sichuan University) and we have been authorized to use it. As your suggestion, we have changed the supporting references. The Chinese version of the QoR-15 is available in Appendix Figure A.

Reference:

“Xue-Shan Bu, Jing Zhang, Yun-Xia Zuo. Validation of the Chinese version of the quality of recovery-15 score and its comparison with the post-operative quality recovery scale. Patient 2016;9(3):251-9. https://doi.org/10.1007/s40271-015-0148-6.”

Comment 5: In line 105, What is the minimum difference? Did you investigate the QoR several times?

Response: We apologize for confusing the reviewer. We assessed QoR-15 only once at the postoperative day 5. We have revised the statement to:

“The difference in the QoR-15 scores on the postoperative day 5 between the high-intensity and non-high intensity groups greater than 8 points was considered clinically meaningful.”

Comment 6: Did all patients visit the hospital 3 months after the surgery? Considering that there are not many excluded patients, I think it would be difficult to investigate for most patients pain score at 3 months after surgery.

Response: Many thanks for your comment. We did not ask patients to come to the hospital for follow-up. Chronic postsurgical pain was collected with a telephone interview at 3 months after surgery. For 3 months follow-up assessments, we would attempt to contact the patient at least three times on 3 separate days. Thus, we collect almost all patient’s (1198/1228, 97.56%) pain assessment at 3 months after surgery. We have provided more details in revised manuscript.

Comment 7: How did you define severe pain in table 3?

Response: Moderate pain was defined as numeric rating scale of 4 to 6, severe pain was 7 to 10. We have revised the statement in method section as below:

“Pain at rest and during movement was assessed at postoperative days 1, 2, 3, and 5 using a scale from 0 to 10, with 0 signifying “no pain” and 10 signifying “extreme pain” (mild, 1 to 3; moderate, 4 to 6; severe, 7 to 10).”

In table 3, we have added a table note.

Round 2

Reviewer 2 Report

Thank you for your efforts.

Author Response

Thank you for considering our manuscript for publication. We highly appreciate the insightful and critical comments from the reviewer.